# Role of Adiponectin and Brain Derived Neurotrophic Factor in Metabolic Regulation Involved in Adiposity and Body Fat Browning

**DOI:** 10.3390/jcm10010056

**Published:** 2020-12-26

**Authors:** Danbi Jo, Yujeong Son, Gwangho Yoon, Juhyun Song, Oh Yoen Kim

**Affiliations:** 1Department of Anatomy, Chonnam National University, Chonnam National University Medical School, Hwasun 58128, Korea; danbijo0818@gmail.com (D.J.); neuroyoon@gmail.com (G.Y.); 2Department of Food Science and Nutrition, Dong-A University, Busan 49315, Korea; ugson95@naver.com; 3Department of Health Science, Graduate School, Dong-A University, Busan 49315, Korea

**Keywords:** adiponectin, brain-derived neurotrophic factor, obesity, fat browning, adipocyte

## Abstract

Obesity, characterized by excessive fat mass, has been emerging as a major global epidemic and contributes to the increased risk of morbidity around the world. Thus, the necessity to find effective therapy and specific regulatory mechanisms is increasing for controlling obesity. Lately, many researchers have been interested in the linkage between obesity and adipokines/myokines, particularly adiponectin and brain-derived neurotrophic factor (BDNF). However, the role of adiponectin and BDNF in adiposity has not been clearly defined yet. We examined the association of adiposity with adiponectin and BDNF through human study (observational study) with Korean women and in vitro experiments. In the human study, we found a negative relationship between adiposity and circulating adiponectins but irregular patterns in the relationship between adiposity and circulating BDNFs. In the in vitro study using 3T3-L1 adipocytes, adiponectin treatment strongly promoted adipocyte differentiation and the fat browning process, whereas BDNF treatment attenuated adipocyte differentiation and the fat browning process in differentiated adipocytes. Our results demonstrate that adiponectin and BDNF play an important role in regulating fat mass and the expression of fat-browning markers in different ways, and also suggest that circulating adiponectin may be used as an important monitoring index for obesity status.

## 1. Introduction

Obesity, characterized by an excess of body-fat mass, has been emerging as a major global epidemic and also contributes to the increased risk of morbidity all over the world [1,2]. According to the World Health Organization (WHO), obese adults are estimated to comprise approximately 39% of the world’s population [3]. Obesity has been known to be influenced by various factors including genetic differences, imbalance in energy intake and expenditure, and lifestyle and environmental changes and has been strongly linked to other disorders such as diabetes, cardiovascular diseases (CVD), cancer, and depression [4,5,6,7,8,9]. Obesity is associated with diverse pathological signs of metabolic disorders including dyslipidemia, high blood pressure, insulin resistance, and inflammatory process, which is involved in the expression of cytokines and adipokines [10,11,12,13,14].

Adipose tissue is known as an endocrine organ to secret various kinds of hormones to affect body metabolisms, and excessive adipose tissues in obese people cause biological metabolic problems [15]. A study reported that highly accumulated adipose tissues promote imbalances in lipid homeostasis and adipokine expressions and subsequently contribute to the poor vasculature in blood vessels, thereby affecting the function of the heart [16]. Another study also demonstrated that excessive adipose tissue overexpresses pro-inflammatory adipokines in the fat, which results in chronic inflammation and ultimately triggers the increase of risk for other diseases such as diabetes and CVD [17].

Among the adipocytes-derived adipokine hormones, adiponectin is well-known for its multifunctional roles due to its anti-diabetic, anti-inflammatory, anti-obesity, and anti-atherosclerosis properties [18,19]. Adiponectin acts by binding specific receptors AdipoR1 and AdipoR2 [20] and activates AMP kinase (AMPK) and peroxisome-proliferator activated receptor-α (PPAR-α) in liver, muscle, and endothelial cells [21] and stimulates Ca^2+^ signaling in cells [22]. Some epidemiological studies proved the potential of adiponectin to reduce the risk of CVD such as obesity, hypertension, diabetes, and atherosclerosis [23,24,25]. A clinical study also mentioned that adiponectin supplementation could reduce the risk of atherosclerosis by inhibiting the secretion of atherogenic factors in blood vessels and blocking the proliferation and migration of vascular smooth muscle cells (VSMCs) [26].

Recent studies highlighted that circulating brain-derived neurotrophic factor (BDNF) in the bloodstream is correlated with metabolic disorders [27,28,29,30] and that metabolic disorders were suggested to have a strongly positive relationship with brain function [31,32,33]. In fact, BDNF, a neurotrophic factor, is mainly secreted from neuronal cells and contributes to the survival ability of neuronal cells, the suppression of neuroinflammation, and the protection of neuronal damage against stress in the central nerve system (CNS), involved in cognitive function [34,35]. BDNF is also expressed in leukocytes, platelets, and endothelial cells and has pivotal roles in the non-neuronal cells [36,37] related to insulin metabolism, lipid metabolisms [29], and cardiovascular homeostasis [35,38,39]. Most BDNFs in the bloodstream are stored in platelets and leukocytes, thereby affecting other tissues [40,41,42]. The alterations of BDNF expression in leukocytes are also associated with diverse psychiatric disorders [43], and changes of BDNF expression in plasma or serum are associated with various neurodegenerative diseases [44]. The current study has shown that BDNF deficiency leads to severe leakage of VSMCs and apoptosis of vascular endothelial cells [37]. In addition, BDNF secretion from monocytes can be stimulated by proinflammatory cytokines, and the elevated plasma BDNF levels indicate an inflammatory state associated with greater adiposity [45]. In fact, BDNF has two isoforms such as mature form (mature-BDNF) and precursor form (proBDNF) [46]. proBDNF is usually converted into mature-BDNF by proteolytical cleavage [47]. Mature-BDNF binds to TrkB receptor [48] and subsequently exerts a physiological change [34,35]. On the other hand, ProBDNF that was not cleaved into mature form induces apoptosis and growth cone retraction through the specific binding to p75(NTR) receptor [49]. Recently, mature-BDNF was reported to play an essential but intricate role in body weight control, and the impairment in the activation of BDNF-TrkB receptor resulted in increased appetite, reduced energy expenditure, and austere obesity [50]. Furthermore, BDNF could be used as a monitoring marker for assessing the process of obesity and the step of visceral adiposity [38,51]. Besides, BDNF has been reported to be positively correlated with numerous metabolic risk factors such as increased circulating levels of triglyceride, glucose, and Hb_A1C_ and insulin resistance [52].

However, the role of adiponectin and BDNF in adiposity has not been clearly defined yet. Therefore, we performed a human study and in vitro experiments to investigate the association between adiposity, and adiponectin and BDNF and the mechanism of how adiponectin and BDNF participate in body fat control.

## 2. Materials and Methods

### 2.1. Study Population

Study subjects were recruited through public advertisement in Busan, South Korea (2018). Korean adult women (19–75 years) who had not been diagnosed with diabetes, stroke, cancer, or other diseases (thyroid, liver, renal disease, etc.) participated in the study. After the screening, a total of 193 Korean adult women were finally included in the analysis. Study women were divided into three groups according to obese status based on the level of body mass index (BMI kg/m^2^): normal weight (BMI: 18.5–22.9, *n* = 97), overweight (BMI 23.0–24.9, *n* = 34), and obesity (BMI ≥ 25.0, *n* = 62) [53]. The study subjects were given an explanation of the study objectives and written consent. The study protocol was approved by the Institutional Review Board in Dong-A University (project identification code: 2-104709-AB-N-01-201603- BR-001-06).

### 2.2. Anthropometric Parameter, Blood pressure, and Blood Collection

Height, weight, body fat percentage, and skeletal muscle mass were measured of subjects wearing light clothes and no shoes and using an automatic body composition analyzer (N20; AIIA Communication Inc., Seongnam, Korea). BMI was calculated as body weight divided by height (kg/m^2^). Waist circumference (WC) was measured in standing subjects after normal exhalation. Systolic and diastolic blood pressures (BPs) were obtained from the left or right arm at the seated using an automatic BP monitor (HEM-7220, Omron, Matsusaka, Japan) after 20 min of rest. After 12 h fast (overnight fasting), venous blood samples were collected in EDTA-treated and plain tubes. Plasma samples were obtained by blood centrifugation at 1300 g for 15 min at room temperature, and serum samples were centrifuged at 2000× *g* for 15 min at room temperature and stored at −80 °C until analysis.

### 2.3. Serum Glycemic Parameters and Lipid Profile

Fasting serum glucose levels were measured using a glucose oxidase method with Beckman Glucose Analyzer (Beckman Ins., Irvine, CA, USA). Hemoglobin A1c (HbA1c) was measured using VARIANT II Turbo HbA1c kit-2.0 (Bio-Rad, Hercules, CA, USA). Insulin and C-peptide levels were measured by radioimmunoassays with commercial kits (ImmunoNucleo Corporation, Stillwater, MN, USA). Homeostasis model assessment insulin resistance (HOMA-IR) was calculated using HOMA developed by Matthews [26]. HOMA-IR = ((fasting insulin (μIU/mL) × fasting glucose (mg/dL))/450. Serum total-cholesterol and triglyceride were measured by enzymatic assay using commercially available kits on a Hitachi 7150 Autoanalyzer (Hitachi Ltd., Tokyo, Japan). After precipitation of chylomicrons with dextran sulfate magnesium, levels of LDL-cholesterol and HDL-cholesterol in the supernatants were analyzed by an enzymatic method.

### 2.4. Serum High-Sensitivity C-Reactive Protein

Serum high-sensitivity C-reactive protein (hs-CRP) was measured with an ADVIA 1650 system (Bayer, Tarrytown, NY, USA) using a commercially available, high-sensitivity CRP-Latex (II) ×2 kit (Seiken Laboratories Ltd., Tokyo, Japan).

### 2.5. Serum Adiponectin and BDNF Concentrations

Plasma adiponectin concentrations were measured using an enzyme immunoassay (R&D Systems, Minneapolis, USA, DRP300). The assays were performed in duplicate using a microplate absorbance reader (Bio-Rad Laboratories, Hercules, CA, USA) set to 450 nm (intra-assay and inter-assay variations were less than 8% and 5%, respectively). Serum levels of mature-BDNF were measured using an ELISA Kit (Aviscera bioscience, Santa Clara, CA, USA, SK00752-01). The assay was performed in duplicate using a microplate absorbance reader (Bio-Rad Laboratories, Hercules, CA, USA) set to 450 nm (intra-assay and inter-assay variations were less than 10% and 6%, respectively).

### 2.6. T3-L1 Cell Culture and Differentiation

Mouse embryonic fibroblasts 3T3-L1 cell line was purchased from the American Type Culture Collection (ATCC; Manassas, VA, USA). The preadipocyte 3T3-L1 cells were cultured in Dulbecco’s Modified Eagle’s Medium (DMEM) with 10% bovine calf serum (BCS) and 100 U/mL penicillin-streptomycin and maintained at 37 °C in a 5% CO₂ incubator. Cells were cultured in 6-well culture plates with media change for 2 days. Upon reaching 100% confluence (Day 0), cells for differentiation were cultured with 0.5 mM 1-methyl-3-isobutyl-xantin (IBMX), 1 μM dexamethasone, 1 μg/mL insulin in DMEM containing 10% FBS (MDI differentiation medium) for 2 days. After 2 days (Day 2), cells were cultured with 1 μg/mL insulin in DMEM containing 10% FBS (Insulin medium) for another 2 days. After 2 days (Day 4), cells maintained in DMEM containing 10% FBS for 4 days and medium was replaced every 2 days until fully differentiated (Day 8). Cells for undifferentiated adipocytes were maintained for the same period without differentiation-inducing drug treatment. Differentiation protocol is shown in Figure 1.

### 2.7. Drug Treatments

7,8-Dihydroxyflavone (7,8-DHF) and recombinant adiponectin (Acrp30) were purchased from Sigma Aldrich (St. Louis, Mo, USA). For drug treatments, 7,8-DHF was diluted using dimethyl sulfoxide (DMSO) and Acrp30 was diluted with sterilized 1×phosphate-buffered saline (PBS). 3T3-L1 adipocytes were treated with 7,8-DHF or Acrp30 in medium at a concentration of 5 μg/mL or 20 ng/mL, respectively, at Day 7.

### 2.8. Oil Red O Staining

On the day cells were fully differentiated, cells were stained with Oil Red O (Sigma- Aldrich, St. Louis, MO, USA) to analyze intracellular lipid accumulation and level of differentiation in adipocytes. 3T3-L1 adipocytes were washed with cold PBS and fixed with 4% formaldehyde at room temperature for 1 h. Cells were washed twice with distilled water (DW) and incubated in 60% isopropanol for 5 min. Fixed cells were stained with 0.6% Oil Red O solution for 1 h at room temperature. After 1 h, cells were washed with 60% Ethanol and DW. Images were captured using Eclipse Ts2 fluorescent microscope (Nikon, Tokyo, Japan) and analyzed for quantification of Oil Red O staining.

### 2.9. Western Blot Analysis

Protein expression was analyzed by Western blot assay. 3T3-L1 adipocytes were collected after full differentiation. Cells were homogenized with ice-cold RIPA buffer (Translab). Protein contents were quantified using a BCA protein assay kit (Thermo Fisher Scientific, Waltham, MA, USA). Protein (15, 25 μg) was separated on 10–12% SDS-polyacrylamide gel and transferred onto methanol-activated polyvinylidene difluoride (PVDF) membrane. Membranes were blocked with 5% skim milk or 5% bovine serum albumin (BSA) in 1×TBS-T (10×TBS buffer, 0.1% Tween-20) for 1 h followed by incubation with specific primary antibody overnight at 4 °C. Primary antibodies: phosphor-AKT (1:1000, cell signaling, residues surrounding Ser473), AKT (1:1000, cell signaling, C-terminal sequence), phosphor-CREB (1:1000, cell signaling, residues surrounding Ser133), CREB (1:1000, Cell signaling, N-terminus), phosphor-ERK1/2 (1:2000, cell signaling, residues surrounding Thr202/Tyr204), ERK1/2 (1:1000, cell signaling, residues near the C-terminus), phosphor-AMPK (1:1000, cell signaling, residues surrounding Thr172), AMPKα (1:1000, cell signaling, N-terminus), phosphor-p38 MAPK (1:1000, cell signaling, residues surrounding Thr180/Tyr182), p38 MAPK (1:1000, cell signaling, C-terminus), PGC-1α (1:1000, Abcam, amino acid 777–797), PPAR γ (1:1000, cell signaling, residues surrounding His494), β-actin (1:1000, cell signaling, N-terminus). Membranes were then incubated with horseradish peroxidase (HRP)-conjugated secondary antibodies for 2 h at room temperature. After incubation, membranes were detected by ECL solution (Thermo Fisher Scientific, Rockford, IL, USA) and Fusion Solo software (Vilber, Marne-la-Vallée, France). Protein expression was quantified using Fusion Solo with expression levels normalized to β-actin. Each experiment was repeated three times.

### 2.10. Quantitative Real-Time PCR

To confirm the effect of 7,8-DHF and Acrp30 on fat browning in 3T3-L1 adipocyte differentiation, quantitative real-time PCR was performed in triplicate using each primer. Total RNA was extracted from differentiated 3T3-L1 adipocytes using TRIzol reagent (Ambion, Ausin, TX, USA). Complementary DNA (cDNA) synthesized using RevertAid reverse transcriptase (Thermo Fisher Scientific, Rockford, IL, USA) from extracted RNA. SYBR green PCR master mix was used with cDNA, and each primer to perform real-time PCR using Step One Plus real-time PCR system (Applied Biosystems, Foster City, CA, USA). PCR was performed using fat-browning-related primers and housekeeping gene glyceraldehyde-3-phosphate dehydrogenase (GAPDH). Fat browning primers are shown in Appendix A. Each experiment was repeated three times.

### 2.11. Statistical Analysis

Statistical analyses were performed using SPSS version 25.0 for Windows (SPSS Inc., Chicago, IL, USA). Differences of parameters between the three groups were tested by one-way analysis of variance (unadjusted) (P0), and general linear model by Bonfferoni correction with adjustment for age, smoking, drinking, menopausal status, and total calorie intake (P1). Differences of parameters between the two groups were tested by the Student *t*-test method. Frequency was tested with the Chi-square test. The distribution of continuous variables was inspected to detect non-normal distribution before statistical analysis. Skewed variables were log-transformed for statistical analysis. For descriptive purposes, the mean values were presented using untransformed values expressed as means ± standard error (S.E) or percentages. To evaluate the association between circulating adiponectin levels and obesity, logistic regression analysis that has two-step models for the odds ratios (ORs, 95% confidence intervals (CIs)) were performed. OR0 was not adjusted. OR1 was adjusted for age, total energy intake, smoking, and drinking habits, and menopausal status. A two-tailed *p*-value of less than 0.05 was considered statistically significant.

## 3. Results

### 3.1. Baseline and Biochemical Characteristics of Study Subjects According to Obesity Status

Firstly, we investigate the basic profiles and biochemical characteristics of 193 study subjects according to obesity status estimated by BMI (Table 1). We found that obese people (BMI >25, *n* = 62) were significantly older than normal-weight (BMI 18.5–22.9, *n* = 97) and overweight (BMI 23.0–24.9, *n* = 34) people. The levels of systolic BP, waist circumference, percentage body fat, C-peptide, and HOMA-IR were highest in the obesity group, and higher in the overweight group than in the normal-weight group. Diastolic BP, heart rate, and circulating levels of insulin, triglyceride, and hs-CRP were higher, but percentage skeletal muscle and HDL-cholesterol levels were lower in the obesity group than the normal-weight and overweight group. LDL-cholesterol levels were significantly higher in overweight and obesity groups than in the normal-weight group. On the other hand, heart rate fasting levels of glucose and total cholesterol and HbA1C percentage were not significantly different among the groups.

### 3.2. Circulating Levels of Adiponectin and Brain-Derived Neurotrophic Factors (BDNFs) According to Obesity Status

Figure 2 presents circulating levels of adiponectin and mature-BDNF according to obesity status. Adiponectin concentrations in obese women (4.06 ± 0.47 μg/mL) were significantly lower than those in normal-weight women (5.63 ± 0.49 μg/mL) after adjustment for age, smoking, drinking, menopausal status, and total calorie intake (*p*1 = 0.021). On the other hand, mature-BDNF levels were significantly lower in overweight women (14.76 ± 1.23 ng/mL) than in normal-weight (15.71 ± 0.74 ng/mL) and obese women (17.08 ± 1.07 ng/mL) (*p*1 = 0.035). There was no significant difference in circulating adiponectin levels between normal-weight and obese women.

### 3.3. Adiposity Estimated by Body Mass Index and Waist Circumference According to Circulating Levels of Adiponectin and Active-BDNF

Based on the results in Figure 2, we subdivided study subjects into quartile groups according to circulating adiponectin or active-BDNF levels (Figure 3A,B). Both BMI and waist circumference levels were significantly lower in the lowest adiponectin group (Q4) than in the highest adiponectin group (Q1) after the adjustment (Figure 3A). On the other hand, the levels of BMI and waist circumference show inconsistent patterns according to mature-BDNF levels (Figure 3B).

### 3.4. Risk of Obesity (BMI ≥25 kg/m^2^) Associated with Circulating Adiponectin Concentrations in Korean Women

Based on the results in Figure 2 and Figure 3, we further investigated the association between circulating adiponectin and obesity by comparison of ORs (95% CIs) calculated with a logistic regression model with adjustment for confounding factors (i.e., age, cigarette smoking, alcohol consumption, menopausal status, and total calorie intake). Risk for obesity (BMI ≥25 kg/m^2^) was significantly lower in the highest adiponectin group (Q4) compared with the lowest adiponectin group (Q1, reference) before and after the adjustment (OR0:0.346; 95%CI, 0.136-0.8812, OR1: 0.249; 95%CI, 0.082-0.757) (Figure 4). On the other hand, no significant association between circulating mature-BDNF and obesity were observed.

### 3.5. The Effect of Adiponectin in 3T3-L1 Adipocyte Cells during Differentiation

To find the mechanism of adiponectin effect on adipocytes during the differentiation process, we conducted several in vitro experiments using 3T3-L1 cell line (Figure 5A–C and Appendix A). When we treated with 20 ng/mL of Acrp30, a globular form of adiponectin in 3T3-L1 cell lines during differentiation, we observed that Acrp30 markedly promoted adipocyte differentiation using Oil red O staining (Figure 5A). Moreover, we measured protein levels of several markers related to adipocyte differentiation including p-AMPK, p-p38MAPK, PGC-1α, and PPAR-γ through Western blotting (Figure 5B and Figure 6). We observed that p-AMPK, p-p38 MAPK, and the expression of PPAR-γ were significantly increased by the treatment with 20 ng/mL of Acrp30 in adipocyte 3T3-L1 cells during differentiation. On the other hand, the protein level of PGC-1α was significantly reduced by the treatment with 20 ng/mL of Acrp30 (Figure 5B and Appendix A). In addition, we conducted quantitative RT-PCR for measuring mRNA levels of fat browning factors (Figure 5C). We found that Acrp30 treatment induced the expression of fat browning factors such as Prdm 16, UCP-1, Cidea, and Elovl3, except Zfp516 in 3T3-L1 cells by the treatment with 20 ng/mL of Acrp30 during differentiation.

### 3.6. The Effect of BDNF in 3T3-L1 Adipocyte Cells During Differentiation

To confirm the mechanism of BDNF’s effect on adipocytes during the differentiation process, we conducted several in vitro experiments using 3T3-L1 cell line (Figure 6A–C and Appendix A). When we treated with 5 μg/mL of 7,8-DHF, a BDNF mimic molecule in 3T3-L1 cell lines during differentiation, we observed that 7,8-DHF treatment considerably reduced adipocyte differentiation using Oil red O staining (Figure 6A). In addition, we measured protein levels of several markers related to adipocyte differentiation including p-AKT, p-CREB, p-ERK1/2, PGC-1α, and PPAR-γ (Figure 6B and Appendix A). Given protein level data using Western blotting, we figured out that p-Akt and p-ERK1/2 and the expression of PPAR-γ were attenuated by the treatment with 5 μg/mL of 7,8-DHF in adipocyte 3T3-L1 cells during differentiation. On the other hand, p-CREB and expression of PGC-1α were not significantly changed (Figure 6B and Appendix A). Our PCR data showed the changes of several fat browning factors in 3T3-L1 cells by the treatment with 5 μg/mL of 7,8-DHF during differentiation (Figure 6C). The mRNA expressions of Prdm16, Cidea, and Elovl3 were significantly reduced in 3T3-L1 cells by the treatment with 5 μg/mL of 7,8-DHF during differentiation. However, the reduced levels of UCP-1 and Zfp516 did not reach statistical significance (Figure 6C).

## 4. Discussion

In this study, we tried to investigate the metabolic regulation, including the fat-browning-related specific mechanism, of adiponectin and BDNF in adiposity through human study and in vitro study. Our human data showed that obese women showed low adiponectin levels compared to normal subjects, and the cases with higher adiponectin levels in circulation had a significantly lower risk of obesity than those with lower adiponectin levels, whereas the levels of mature-BDNFs were not consistent by obese status. From the in vitro data, we found interesting results that adiponectin (Acrp30) treatment dramatically promoted 3T3-L1 adipocyte differentiation but induced the expression of fat-browning factors. On the other hand, BDNF (7,8-DHF) treatment markedly attenuated adipocytes differentiation and the expression of fat-browning-related factors as well. Our results demonstrated that adiponectin and BDNF play important roles in regulating fat mass and the expression of fat-browning markers in different ways. We assumed that adiponectin promotes both adipocyte differentiation and fat-browning process, and BDNF inhibits adipocyte differentiation, ultimately leading to the reduction of BMI and fat mass and protecting fat accumulation. We also suggested that circulating adiponectin may be used as an important monitoring index for obesity status.

Adiponectin is an adipocyte-secreted adipokine that has multifunctional roles related to anti-diabetic, anti-inflammatory, anti-obesity, and anti-atherosclerotic activities and related pathologies [18,19]. Adiponectin circulates from 5 to 30 mg in human blood [18], and the gene coding adiponectin located on chromosome 3q27 in humans is associated with metabolic diseases such as diabetes and atherosclerosis [54]. Adiponectin is known to activate AMPK, and PPAR-α in liver, muscle, and endothelial cells [21] and stimulates Ca^2+^ signaling in cells [22]. It also suppresses the production of reactive oxygen species (ROS) [55], oxidized LDL-cholesterol [56], and palmitate [57] mediated by cAMP-dependent protein kinase A [55] and AMPK [57,58].

Previous epidemiological studies demonstrated that adiponectin might reduce the risk of CVD (i.e., obesity, hypertension, diabetes, and atherosclerosis) [23,24,25]. Other studies reported that changes in adiponectin levels in postmenopausal women were dependent of aging and low sex-hormone-binding globulin [59,60]. It was also reported that the risk of atherosclerosis was reduced by adiponectin supplementation, which inhibited the secretion of atherogenic factors in blood vessels and the proliferation and migration of VSMCs [26].

Our data showed increases of p-AMPK, p-p38MAPK, and PPARγ expression in adipocyte 3T3-L1 cell lines during differentiation, and higher adiponectin levels in circulation were negatively associated with increased BMI and adiposity in study subjects. These results indicate that adiponectin could accelerate adipocyte differentiation. Our PCR data also indicated that adiponectin induced the turn-on of the fat-browning process, leading to high calorie expenditure and active burning of fat mass in body. The modulation of adiponectin function to promote the fat-browning process may be a key to regulate obesity status. In addition, circulating adiponectin levels represent adiposity status in the human body

Previous studies have reported that circulating BDNF levels were decreased significantly with increased age [61,62,63]. However, a clinical study showed contradictory results according to age groups [64]. BDNF concentration was positively correlated with age in the middle-aged (<65 years), but negatively correlated in the elderly people (≥65 years). Circulating BDNF levels were also reported to be related to adiposity and were suggested as a monitoring marker for assessing the step of visceral adiposity [38,51]. For example, plasma BDNF levels were correlated positively with BMI and fat mass in women, but this was not observed in men [38]. Serum BDNF level was increased and associated with obesity in women with newly diagnosed type 2 diabetes mellitus [65]. The animal experiments demonstrated that reduced expression of BDNF with the high-fat diet increased the drive to eat, thereby contributing to diet-induced obesity in male rats, but it was not observed in female rats [66]. They also showed that the expression of BDNF mRNA in the ventromedial nucleus of the hypothalamus was more stable in female rats than in male rats even when energy homeostasis was disturbed. This may suggest sex-distinct regulation of central BDNF expression by diet and energy status [66]. However, our data showed that circulating mature-BDNF levels were inconsistent according to obesity status of the study women. We assumed that the discrepancy between our results and those of previous studies may be related to the degree of obesity and the baseline health status. Our study used the Asian criteria for obesity (BMI ≥25 kg/m^2^) instead of the Western criteria (BMI ≥30 kg/m^2^) and included relatively healthy subjects without diagnosed disease. Therefore, our study subjects were less obese and showed no severe metabolic abnormality compared with obese people by the Western criteria. In the future, we shall include more obese people according to the Western criteria. In addition, the results from in vitro experiments show that BDNF treatment basically attenuated adipocyte differentiation, but did not promote the body-fat-browning process under differentiated adipocytes. The results were distinctly different from those observed when Acrp30 (adiponectin) was treated. This may be partly explained by the expression data from 3T3-L1 adipogensis presented in the GEO dataset showing that BDNF levels were high before 3T3-L1 differentiation and then constantly decreased during adipogenesis [67]. This is why BDNF treatment inhibited adipocyte differentiation but did not promote a fat-browning process after the adipocyte differentiation. In the future, we need to confirm the role of BDNF in adipogensis and the fat-browning process using primary adipocytes.

Regarding the relationship between BDNF and adiposity based on our human data and in vitro data, we assume that BDNF may contribute to the attenuation of adipocyte differentiation, but its regulatory effect such as body fat-browning under metabolic imbalance status (i.e., adiposity and obesity) was differently observed. Recent studies reported the relationships between circulating BDNF and metabolic disorders [27,28,29,30], which is closely associated with brain function [31,32,33]. BDNF is a well-known neurotrophic factor mainly secreted from neuronal cells [34,35] but also plays an important role in non-neuronal cells [36,37] in relation with lipid metabolism [29], visceral obesity, insulin regulation, and cardiovascular homeostasis [35,38,39,51]. Previous studies also mentioned that BDNF is involved in atherogenesis and blood plaque formation in blood vessels by controlling the activation of NADPH oxidase [68] and that it inhibits cardiac infarction and prevents cardiac pathogenesis [69]. Kaess et al. mentioned that low levels of BDNF lead to the high incidence of CVD by aggravating vascular function [70]. On the other hand, BDNF was positively correlated with numerous metabolic risk factors (i.e., increased triglyceride, glucose, and Hb_A1C_, as well as insulin resistance, in the bloodstream) [52]. Based on previous significant findings, BDNF level is considerably linked to the alteration of metabolic risk factors leading to metabolic disorders such as obesity, diabetes, CVD, and CNS diseases [71]. Lately, the study of adiponectin and BDNF focuses on the finding of common therapy between metabolic disorders and CNS disease and emphasizes a greater understanding of the common regulatory mechanisms both in metabolic syndromes and brain dysfunction [72]. Given previous significant studies and our results, we assume that BDNF has a different role in adipocytes under metabolic imbalance such as obese states, differently from the effect of adiponectin on body fat control.

Our study has limitations. First, we included only women subjects without diagnosed diseases in the human study, because previous studies reported that circulating levels of adiponectin and BDNF were usually higher in women than in men, and their relationship with anthropometric and metabolic parameters was differently observed between men and women [38,73,74]. Moreover, women might be in relatively simple conditions such as less exposure to cigarette smoking and heavy alcohol drinking than men, even though menopausal status still affect the metabolic condition. Therefore, we adjusted several potential confounding factors including age, cigarette smoking, alcohol consumption, calorie intake, and menopausal status for the statistical test. Second, we used the Asian criteria for obesity categorization (BMI ≥25 kg/m^2^). Based on the Western criteria (BMI ≥30 kg/m^2^), only 8 of 193 study subjects were obese in this study. Third, we performed the study in a cross-sectional way; thus, we could not identify the causal effect of circulating adiponectin and mature BDNF on the adiposity change as well as the expression of fat-browning-related markers by the weight control. Fourth, we used the 3T3-L1 cells, not the primary adipocytes for the in vitro experiments, even though we referred to previous reports that 3T3-L1 preadipocyte had been used for observing phenotype changes related to fat browning and adipogenesis [75,76,77]. In the future, we need to include both men and women, use Western criteria for obesity after including a larger population, and perform the clinical trial for weight control. In addition, we need to use primary adipocytes in the in vitro experiments to more clearly support the study conclusion for the role of adiponectin and BDNF in brown adipocyte functions.

Despite the limitations, we demonstrated the role of adiponectin and BDNF in the regulation of adiposity and the related mechanisms through human study and in vitro study. We also propose that adiponectin may play an important role in promoting both adipocyte differentiation and fat-browning process, thereby contributing to the reduction of fat mass and protection of fat accumulation in the body. On the other hand, mature-BDNF may differently contribute to body fat regulation according to obese state. Thus, further study about the roles of adiponectin and BDNF in obese status is essential for finding the regulatory mechanisms and therapeutic solution for obesity.

## Figures and Tables

**Figure 1 jcm-10-00056-f001:**
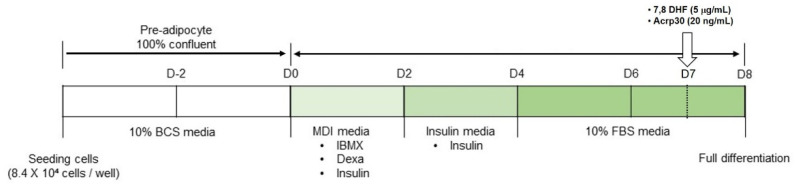
Differentiation protocol of 3T3-L1 cells. BCS: bovine calf serum, Dexa: dexamethasone, DMEM: Dulbecco’s Modified Eagle’s Medium, FBS: fetal bovine serum, IBMX: 1-methyl-3-isobutyl-xantin.

**Figure 2 jcm-10-00056-f002:**
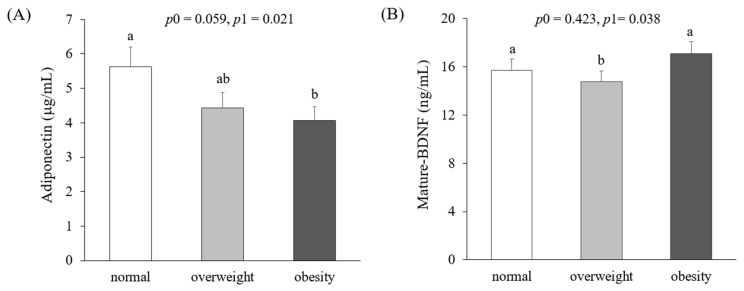
Circulating levels of adiponectin (**A**) and mature-brain-derived neurotrophic factors (BDNFs) (**B**) according to obesity status. Means ± S.E. tested by one-way analysis of variance (ANOVA) (*p*0) and general linear model by Bonfferoni correction with adjustment for age, smoking, drinking, menopausal status, and total calorie intake (*p*1). Sharing the same alphabet letter (a or b) indicates no statistical significance among the values.

**Figure 3 jcm-10-00056-f003:**
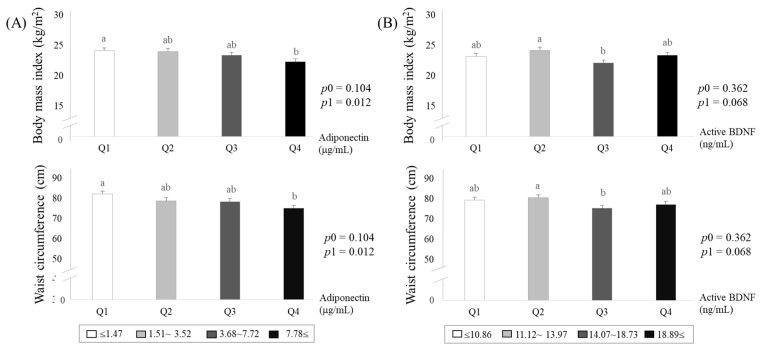
Adiposity estimated by body mass index and waist circumference according to circulating levels of adiponectin (**A**) and mature-brain-derived neurotrophic factors (BDNF) (**B**). Means ± S.E. tested by one-way analysis of variance (ANOVA) (*p*0), and general linear model by Bonfferoni correction with adjustment for age, smoking, drinking menopausal status, and total calorie intake (*p*1). Sharing the same alphabet letter (a or b) indicates no statistical significance among the values.

**Figure 4 jcm-10-00056-f004:**
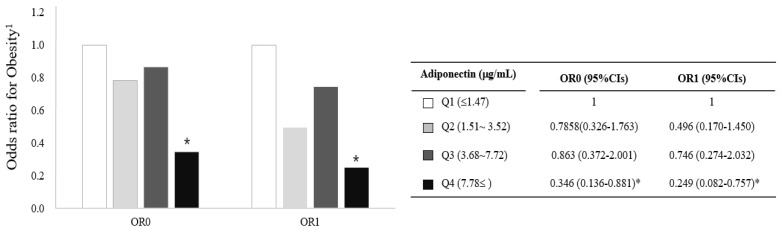
Risk of obesity (BMI ≥25 kg/m^2^) associated with circulating adiponectin concentrations in Korean women. * *p* < 0.05 compared with the reference group (Q1). The association was calculated using the OR (95% CIs) of a logistic regression model with adjustment for age, smoking, drinking, menopausal status, and total calorie intake; ^1^: BMI ≥25 kg/m^2^, CI: confidence interval, OR: odds ratio, Q: quartile.

**Figure 5 jcm-10-00056-f005:**
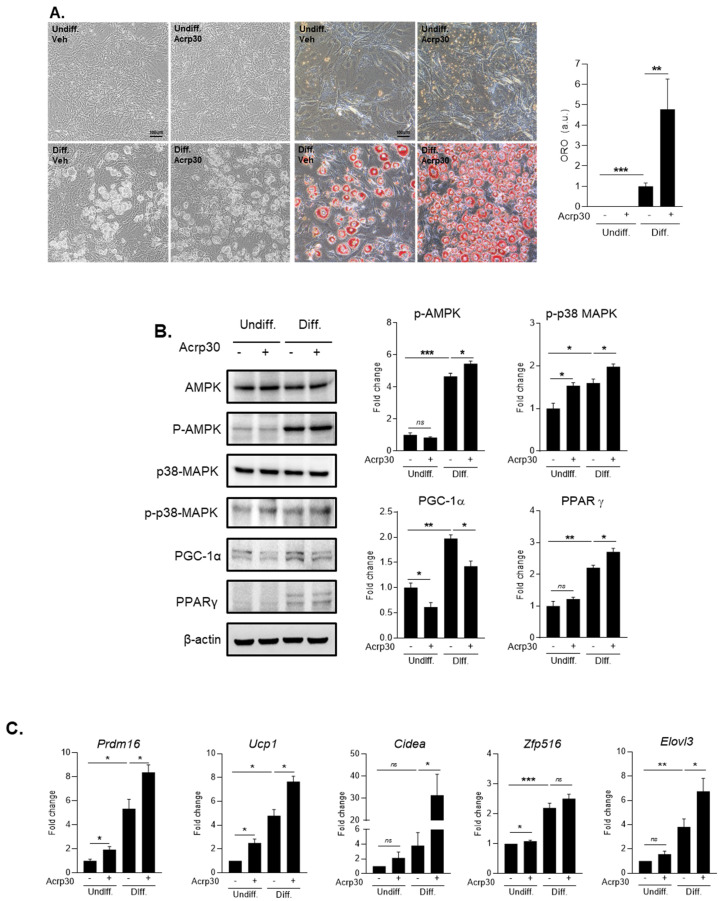
The effect of Acrp30 (adiponectin) treatment in undifferentiated and differentiated 3T3-L1 adipocyte cells. (**A**) Adipocyte differentiation and lipid accumulation by Oil red O staining, (**B**) protein expression related to adipocyte differentiation, (**C**) mRNA expression of genes involved in body fat browning. Means ± S.E. tested by independent *t*-test (nonparametric). * *p* < 0.05, ** *p* < 0.01, *** *p* < 0.001, n.s.: no significance; 20 ng/mL treatment of Arp30; undiff: undifferentiated cells, diff: differentiated cells.

**Figure 6 jcm-10-00056-f006:**
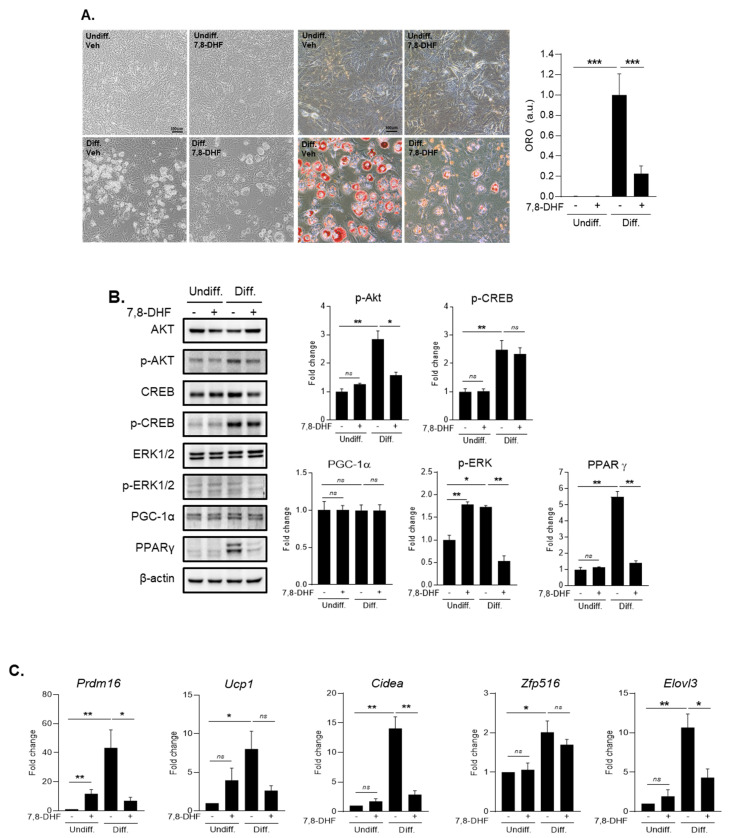
The effect of 7,8-DHF (BDNF mimetic) treatment in undifferentiated and differentiated 3T3-L1 adipocyte **cells.** (**A**) adipocyte differentiation and lipid accumulation by Oil red O staining, (**B**) protein expression related to adipocyte differentiation, (**C**) mRNA expression of genes involved in body fat-browning. Means ± S.E. tested by independent *t*-test (nonparametric). * *p* < 0.05, ** *p* < 0.01, *** *p* < 0.001, n.s.: no significance; 5 μg/mL treatment of 7,8-DHF; undiff: undifferentiated cells, diff: differentiated cells.

**Table 1 jcm-10-00056-t001:** Anthropometric and biochemical characteristics of study women according to obesity status.

Variables	Normal Weight (BMI 18.5–22.9) *n* = 97	Overweight (BMI 23.0–24.9) *n* = 34	Obesity (BMI ≥25.0) *n* = 62	*p*-Value
Age (years)	43.5 ± 1.22 ^b^	44.7 ± 2.31 ^b^	50.7 ± 1.46 ^a^	0.001
Body weight (kg)	53.6 ± 0.46 ^c^	60.1 ± 0.90 ^b^	68.3 ± 0.83 ^a^	<0.001
Body mass index (kg/m^2^)	21.0 ± 0.12 ^c^	23.8 ± 0.11 ^b^	27.6 ± 0.29 ^a^	<0.001
Systolic BP (mmHg)	110.4 ± 1.01 ^c^	114.2 ± 1.82 ^b^	122.2 ± 1.82 ^a^	<0.001
Diastolic BP (mmHg)	71.3 ± 0.79 ^b^	72.2 ± 1.30 ^b^	77.1 ± 1.11 ^a^	<0.001
Heart rate	67.9 ± 0.99	68.9 ± 1.51	71.6 ± 1.30	0.069
Waist circumference (cm)	72.7 ± 0.61 ^c^	79.3 ± 0.93 ^b^	88.8 ± 0.96 ^a^	<0.001
Body fat (%)	28.0 ± 0.90 ^c^	33.2 ± 1.32 ^b^	37.9 ± 0.78 ^a^	<0.001
Skeletal muscle (%)	38.2 ± 0.57 ^a^	37.1 ± 1.75 ^a^	33.5 ± 0.48 ^b^	<0.001
Glucose (mg/dL) ^§^	88.5 ± 1.25	89.3 ± 2.51	93.0 ± 2.07	0.105
HbA1c %^§^	5.36 ± 0.05	5.38 ± 0.08	5.50 ± 0.04	0.084
Insulin (μIU/mL)	7.94 ± 1.10 ^b^	9.79 ± 2.55 ^b^	13.1 ± 1.69 ^a^	0.041
C-peptide (ng/mL) ^§^	1.62 ± 0.14 ^c^	2.24 ± 0.36 ^b^	2.70 ± 0.27 ^a^	0.004
HOMA-IR ^§^	1.76 ± 0.27 ^c^	2.30 ± 0.66 ^b^	3.38 ± 0.62 ^a^	<0.001
Triglyceride (mg/dL) ^§^	85.8 ± 5.88 ^b^	66.9 ± 4.45 ^c^	112.3 ± 7.12 ^a^	<0.001
HDL-C (mg/dL)	65.2 ± 1.55 ^a^	66.6 ± 2.45 ^a^	59.3 ± 1.79 ^b^	0.021
LDL-C (mg/dL) ^§^	114.9 ± 3.05 ^b^	124.7 ± 5.85 ^a^	128.4 ± 4.04 ^a^	0.022
Total-C (mg/dL) ^§^	188.2 ± 3.0	196.3 ± 6.21	199.4 ± 4.46	0.124
hs-CRP (mg/dL) ^§^	0.58 ± 0.14 ^b^	0.38 ± 0.06 ^b^	1.89 ± 0.64 ^a^	<0.001

Means ± S.E., tested by one-way ANOVA, ^§^ tested after log-transformation; *p* < 0.05 indicates significant differences among the values for the same variable. Sharing the same alphabet letter (a, b, or c) indicates no significant difference among the values in the same row. BMI: body mass index, BP: blood pressure, HOMA-IR: Homeostatic model assessment for insulin resistance, HDL-C: high-density lipoprotein cholesterol, LDL-C: low-density lipoprotein cholesterol, Total-C: total cholesterol, hs-CRP: high sensitivity C-reactive protein.

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
