# Peer review of "Role of Adiponectin and Brain Derived Neurotrophic Factor in Metabolic Regulation Involved in Adiposity and Body Fat Browning"

_jcm, 2020, doi:10.3390/jcm10010056_

Round 1

Reviewer 1 Report

The present paper focuses on the role of adiponectin and BDNF during adipogenesis (in vitro studies on 3T3 cells) and in overweight and obese Korean women.

The study is well designed in its totality.

However, I want to rise a few points:

-in the introduction and the discussion, the authors well discuss the role of BDNF in the central nervous system and peripheral tissues. However, they never mention the important contribution of platelets and leukocytes as the most important reservoir of BDNF in the bloodstream (10.5498/wjp.v6.i1.84) This is important to describe since the authors measured circulating BDNF levels in serum, in which an important contribution of platelets and leukocytes has been demonstrated (10.3390/ijms18091987).

-the authors used the molecule 7,8 DHF as an analog of BDNF to conduct their experiments. Why did they not use human recombinant BDNF protein that is commercially available? In particular, how did they decided the amount of the analog molecule 7,8-DHF, and how it can be compared in terms of potency to BDNF?

-the authors used the analog 7,8-DHF in 3T3 cells already differentiated to adipocytes to clarify the role of BDNF in adipogenesis and browning. They should mention in my opinion this GEO dataset (https://www.ncbi.nlm.nih.gov/geo/query/acc.cgi?acc=GSE6794) in which it was shown that BDNF levels are high before 3T3 differentiation and then they constantly decrease during adipogenesis. In line with this previous data, the negative role of BDNF in already differentiated 3T3 cells during differentiation and browning showed by the authors could be expected, since adding 7,8-DHF they put the cells in a nonphysiological condition.

Author Response

Answers for Reviews’ comments

Manuscript ID: JCM-1050424.R1

Title: Role of Adiponectin and Brain Derived Neurotrophic Factor in Metabolic Regulation Involved in Adiposity and Body Fat Browning

Journal of Clinical Medicine

Dear Reviewer #1

We sincerely appreciate the time spent in reviewing this manuscript and your advice to improve it.

Please, see below our answers to your queries and comments. We also marked the corrected and revised parts of the text with red. We hope that you find them satisfactory.

Reviewer 1’s comments: The present paper focuses on the role of adiponectin and BDNF during adipogenesis (in vitro studies on 3T3 cells) and in overweight and obese Korean women. The study is well designed in its totality. However, I want to rise a few points:

1) In the introduction and the discussion, the authors well discuss the role of BDNF in the central nervous system and peripheral tissues. However, they never mention the important contribution of platelets and leukocytes as the most important reservoir of BDNF in the bloodstream (10.5498/wjp.v6.i1.84). This is important to describe since the authors measured circulating BDNF levels in serum, in which an important contribution of platelets and leukocytes has been demonstrated (10.3390/ijms18091987).

Answer) The authors sincerely appreciate the reviewer’s comment and advice for improving the manuscript. In accordance with your comments, we mentioned the important contribution of platelets and leukocytes as the most important reservoir of BDNF in the blood stream in the introduction and the discussion sections. 

“…… BDNF is also expressed in leukocytes, platelets, and endothelial cells, and has pivotal roles in the non-neuronal cells [36, 37], related with insulin metabolism, lipid metabolisms [29], and cardiovascular homeostasis [35, 38, 39]. Most of BDNFs in blood stream are stored in platelets and leukocytes, thereby affecting other tissues [10.5498/wjp.v6.i1.84, 10.3389/fimmu.2020.575607, 10.1055/s-0037-1613072]. The alterations of BDNF expression in leukocytes are also associated with diverse psychiatric disorders [10.1038/tp.2016.214], and changes of BDNF expression in plasma or serum are associated with various neurodegenerative diseases [10.3390/ijms18091987]. Current study have shown that BDNF deficiency leads to severe leakage of VSMCs and apoptosis of vascular endothelial cells [37]. In addition, BDNF secretion from monocytes can be stimulated by proinflammatory cytokines, and the elevated plasma BDNF levels indicates an inflammatory state associated with greater adiposity [10.1084/jem.189.5.865]. ……. ”

2) the authors used the molecule 7,8 DHF as an analog of BDNF to conduct their experiments. Why did they not use human recombinant BDNF protein that is commercially available? In particular, how did they decide the amount of the analog molecule 7,8-DHF, and how it can be compared in terms of potency to BDNF?

Answer) As you commented, we used the molecule 7,8-DHF as a mimic of BDNF in our experiments. Human recombinant BDNF used in clinical practice has short half-life (only 1-10 minutes) in the plasma [10.1016/0169-328x(95)00250-v]. The 7,8-DHF binds to the TrkB receptor and subsequently exerts a physiological action [10.1073/pnas.1718683115] and prevents the increase of p75(NTR) receptor, thereby ameliorating the imbalance of p75NTR/TrkB [10.1093/hmg/ddx198]. The 7,8-DHF was also reported to have anti-obesity effect and have no detectable toxicity with chronic treatment [10.1073/pnas.1718683115]. Furthermore, it has an inhibitory effect of adipogenesis in 3T3-L1 preadipocyte cells. Based these evidences, we used the 7,8-DHF as a BDNF mimic molecule in this study.

In addition, We used 20 mM (5 mg/mL) of 7,8-DHF in the experiment based on previous paper [10.1016/j.lfs.2015.11.028]. Moreover, we found that the degree of lipid accumulation in 3T3-L1 preadipocyte cells decreased significantly at 20uM (5 mg/mL) through our experiments.

3) the authors used the analog 7,8-DHF in 3T3 cells already differentiated to adipocytes to clarify the role of BDNF in adipogenesis and browning. They should mention in my opinion this GEO dataset (https://www.ncbi.nlm.nih.gov/geo/query/acc.cgi?acc=GSE6794) in which it was shown that BDNF levels are high before 3T3 differentiation and then they constantly decrease during adipogenesis. In line with this previous data, the negative role of BDNF in already differentiated 3T3 cells during differentiation and browning showed by the authors could be expected, since adding 7,8-DHF they put the cells in a nonphysiological condition.

Answer) The authors sincerely appreciate the reviewer’s comment for improving this manuscript. As you commented, our results might be expected by the expression data from 3T3L-1 adipogensis shown in the GEO dataset. The reason why we put the 7,8-DHF in the differentiated adipocyte is to observe if BDNF play a role in adipogenesis and fat-browning during adipocyte differentiation as mentioned in the method and result section,. In accordance with your advice, we mentioned the GEO data set for explaining our results in the discussion part (https://www.ncbi.nlm.nih.gov/geo/query/acc.cgi?acc=GSE6794).

Reviewer 2 Report

Obesity and related metabolic diseases are increasing worldwide. Understanding the association between adipokines and other circulating factors would contribute to the prevention and treatment of obesity. This manuscript entitled “Role of Adiponectin and Brain Derived Neurotrophic Factor in Metabolic Regulation Involved in Adiposity and Body Fat Browning” and authored by Jo et al. investigated an anti-obesity adipokine adiponectin (also anti-metabolic disorders) and brain-derived neurotrophic factor (BDNF) that is associated to metabolic disorders, first a human observational study with women with different BMI, then an in vitro study using 3T3-L1 adipocytes. The authors reported different effects of adiponectin and mature BDNF in regulating fat development, with adiponectin being a better index for obesity. This topic is interesting and important in the field. Below concerns and comments need to be addressed.

1. Introduction:

Below contents are needed in the Introduction:

Are receptors of adiponectin and BDNF presented in adipose tissues and 3T3-L1 adipocytes? What are different forms of BDNF, and what’s the rationale for studying the mature BDNF only. The human study included only women; thus, the authors need to discuss if there are sex differences in the effects of adiponectin and BDNF in obesity and metabolic diseases.

Lines 46-47: need to revise this sentence, as not all adipokines “results in chronic inflammation, and ultimately triggers the increase of risk for other diseases such as diabetes and CVD”.

Line 59: “BDNF is … suggested strong positive relationship between metabolic disorders …” Does this sentence mean that BDNF increases risks for metabolic disorders?

2. Methods:

2.1 What’s the age range of subjects in the human study?

Obesity is usually defined as BMI > 30 and overweight BMI between 25 and 30. I understand that Asian women would have different diagnostic criteria as women in western countries. If there were any subjects with BMI over 30, authors should provide data analysis using western criteria?

2.5 It is unclear that the intra-assay and inter-assay variations were for adiponectin or BDNF assay? The authors should provide this information for both assays.

2.6 Ideally is to use primary adipocytes for the in vitro experiments, to align with the in vivo human study. Additionally, because both adiponectin and BDNF have been reported to induce brown adipocyte function, it is ideal to study brown adipocytes in addition to 3T3-L1 white adipocytes. Such consideration needs to be included into the manuscript.

The study used insulin in the cell culture medium. Is this a standard procedure? Does insulin have crossover action as BDNF (a growth factor)?

2.7 The dosages of Acrp30 (20 ng/ml) and DHF (20 mM) need to be justified. BDNF serum concentration uses unit ng/mL. The authors should use one set of units in one manuscript.

2.9 Does the Western blot antibody for BDNF label mature BDNF only?

2.11 The statistical analysis method listed in this section (t-test) is not consistent with the method list in figure legend (one-way ANOVA). Need to be consistent.

3. Discussion:

BDNF levels were not different among three cohorts of women. Are BDNF expression (gene and protein) and circulating levels changed by body fat in men and women and male and female animal models? (ref: DOI: 10.1016/j.physbeh.2014.03.028)

Line 323: The author discussed adiponectin level affected by aging and presumably low estrogen hormone. Is BDNF regulated by aging and/or estrogen?

Line 346-347: “BDNF level is considerably linked to the alteration of metabolic risk factors, leading to metabolic disorder such as obesity, diabetes, CVD and CNS diseases”. BDNF has been suggested to suppress appetite and increase energy expenditure, thus reduce metabolic risks, which is opposite to this statement. How do the authors reconcile inconsistent literature? How does aging interact with BDNF’s effects in metabolic regulation? Again, the ages of woman subjects in the human study should be included in the manuscript.

Author Response

Answers for Reviews’ comments

Manuscript ID: JCM-1050424.R1

Title: Role of Adiponectin and Brain Derived Neurotrophic Factor in Metabolic Regulation Involved in Adiposity and Body Fat Browning

Journal of Clinical Medicine

Dear Reviewer #2

We sincerely appreciate the time spent in reviewing this manuscript and your advice to improve it.

Please, see below our answers to your queries and comments. We also marked the corrected and revised parts of the text with red. We hope that you find them satisfactory.

Reviewer #2’s Comments: Obesity and related metabolic diseases are increasing worldwide. Understanding the association between adipokines and other circulating factors would contribute to the prevention and treatment of obesity. This manuscript entitled “Role of Adiponectin and Brain Derived Neurotrophic Factor in Metabolic Regulation Involved in Adiposity and Body Fat Browning” and authored by Jo et al. investigated an anti-obesity adipokine adiponectin (also anti-metabolic disorders) and brain-derived neurotrophic factor (BDNF) that is associated to metabolic disorders, first a human observational study with women with different BMI, then an in vitro study using 3T3-L1 adipocytes. The authors reported different effects of adiponectin and mature BDNF in regulating fat development, with adiponectin being a better index for obesity. This topic is interesting and important in the field. Below concerns and comments need to be addressed.

  1. Introduction: Below contents are needed in the Introduction:

Are receptors of adiponectin and BDNF presented in adipose tissues and 3T3-L1 adipocytes? What are different forms of BDNF, and what’s the rationale for studying the mature BDNF only. The human study included only women; thus, the authors need to discuss if there are sex differences in the effects of adiponectin and BDNF in obesity and metabolic diseases.

  Answer) Yes, the 3T3-L1 cells have receptors for adiponectin and BDNF. Adiponectin binds to adiponectin receptors 1 (AdipoR1) and 2 (AdipoR2) [10.1038/s42003-020-01160-4], and 7,8-DHF as a BNDF mimic molecule specifically binds to TrkB receptor [10.3389/fncel.2019.00368] in the 3T3-L1 cell line.

As you commented, BDNF has two isoforms such as mature form (mature-BDNF) and precursor form (proBDNF) [10.3389/fncel.2019.00384]. proBDNF is usually converted into mature-BDNF by proteolytically cleavage [10.1523/JNEUROSCI.0163-13.2013]. Mature BDNF binds to TrkB receptor [10.3892/or.2013.2746] and subsequently exerts a physiological action such as the survival ability of neuronal cells, the suppression of neuroinflammation, and the protection of neuronal damage against stress in central nerve system (CNS), involving in cognitive function [34, 35]. On the other hand, ProBDNF which was not cleaved into mature form induces apoptosis and growth cone retraction through the specific binding to p75(NTR) receptor [10.1186/s13041-018-0411-6]. Recently, mature-BDNF was reported to play an essential, but intricate role in body weight control, and the impairment in the activation of BDNF-TrkB receptor resulted in increased appetite, reduced energy expenditure and austere obesity [10.1016/j.obmed.2020.100189]. Therefore, we investigated the role of mature BDNF in adiposity including adipogenesis and fat-browning.

In addition, as you pointed out, we included only women subjects in the human study. Previous studies reported sex differences in circulating levels of adiponectin and BDNF, that is, adiponectin and BDNF levels in blood stream were significantly higher in women than in men [10.1038/sj.ijo.0803427, 10.1177/0004563217699233, 38]. Serum BDNF level was also found increased and associated with obesity in women with newly diagnosed type 2 diabetes mellitus [10.1016/j.metabol.2006.02.012]. Furthermore, plasma BDNF levels were correlated positively with BMI and fat mass in women, but not in men [38]. Precisely, circulating BDNF levels were positively correlated with BMI, fat mass, diastolic blood pressure, total cholesterol, and LDL-cholesterol, and inversely correlated with folate in females. Circulating BDNF levels were positively correlated with diastolic blood pressure, triglycerides, free thiiodo-thyronine, and bioavailable testosterone, and inversely correlated with sex-hormone binding globulin, and adiponectin in males[38].

Lines 46-47: need to revise this sentence, as not all adipokines “results in chronic inflammation, and ultimately triggers the increase of risk for other diseases such as diabetes and CVD”.

 Answer) As you commented, we revised the sentence.

“Another study also demonstrated that excessive adipose tissue overexpresses pro-inflammatory adipokines in the fat, which results in chronic inflammation, and ultimately triggers the increase of risk for other diseases such as diabetes and CVD.”

Line 59: “BDNF is … suggested strong positive relationship between metabolic disorders …” Does this sentence mean that BDNF increases risks for metabolic disorders?

Answer) We are sorry for making the reviewer confused. We revised the sentence to clarify what we explained.

”Recent studies highlighted that circulating brain-derived neurotrophic factor (BDNF) in blood stream is correlated with metabolic disorders [27-30], and that metabolic disorders were suggested to have strongly positive relationship with brain function [31-33].”

  1. Methods:

2.1 What’s the age range of subjects in the human study? Obesity is usually defined as BMI > 30 and overweight BMI between 25 and 30. I understand that Asian women would have different diagnostic criteria as women in western countries. If there were any subjects with BMI over 30, authors should provide data analysis using western criteria?

Answer) The age range of study subjects were from 19 to 75 years old. Therefore, we compared the parameters among the three groups with the adjustment for age and menopausal status together with smoking and alcohol drinking.

As you commented, we used the Asian criteria for obesity categorization (normal weight: BMI 18.5-22.9; overweight: BMI 23.0-24.9; Obesity: BMI ≥25.0). According to your advice, we categorized them based on the Western criteria (normal weight: BMI 18.5-24.9; overweight: BMI 25.0-29.9; Obesity: BMI ≥30.0).

- by the Asian criteria,

: normal (BMI 18.5-22.9), n=97; overweight (BMI 23.0-24.9), n=34, Obesity (BMI ≥25.0), n=62

- by the Western criteria,

: normal (BMI 18.5-24.9), n=131, overweight (BMI 25.0-29.9), n=54; Obesity (BMI ≥30.0), n=8

We found that only 8 people were obese among 193 of study participants based on the Western guideline. When we subdivided obesity group (BMI ≥25.0) (n=62) into two groups (BMI 25.0-29.9, n=54; BMI ≥30.0, n=8), we still observed the similarly increasing or decreasing patterns in anthropometric and biochemical parameters. Therefore, in this paper, we presented the results by obesity status based on the Asian criteria, and suggested the necessity of further study with large scaled population and categorization of obesity by the Western criteria in the discussion part.

Table . Anthropometric and biochemical characteristics of study women according to BMI status

Variables

BMI 18.5-22.9

(n=97)

BMI 23.0-24.9

(n=34)

BMI 25.0-29.9

(n=54)

BMI ≥30.0

(n=8)

p-value

Age (years)

43.5

±

1.22

44.7

±

2.31

51.22

±

1.56

47.50

±

4.25

0.003

Body weight (kg)

53.6

±

0.46

60.1

±

0.90

66.75

±

0.72

78.39

±

1.86

<0.001

Body mass index (kg/m2)

21

±

0.12

23.8

±

0.11

26.82

±

0.17

32.48

±

0.60

<0.001

Systolic BP (mmHg)

110.4

±

1.01

114.2

±

1.82

121.52

±

2.04

126.75

±

2.86

<0.001

Diastolic BP (mmHg)

71.3

±

0.79

72.2

±

1.30

76.37

±

1.21

82.25

±

2.10

<0.001

Heart rate

67.9

±

0.99

68.9

±

1.51

70.08

±

1.31

82.29

±

2.52

0.001

Waist circumference (cm)

72.7

±

0.61

79.3

±

0.93

87.63

±

0.88

97.87

±

3.57

<0.001

Body fat (%)

28

±

0.90

33.2

±

1.32

36.79

±

0.65

45.96

±

2.01

<0.001

Skeletal muscle (%)

38.2

±

0.57

37.1

±

1.75

34.19

±

0.41

28.86

±

1.26

<0.001

Glucose (mg/dL)§

88.5

±

1.25

89.3

±

2.51

93.74

±

2.32

88.13

±

3.29

0.135

HbA1c %§

5.36

±

0.05

5.38

±

0.08

5.51

±

0.05

5.45

±

0.06

0.168

Insulin (μIU/mL)

7.94

±

1.10

9.79

±

2.55

13.06

±

1.91

13.72

±

2.18

0.094

C-peptide (ng/mL)§

1.62

±

0.14

2.24

±

0.36

2.71

±

0.31

2.58

±

0.31

0.012

HOMA-IR§

1.76

±

0.27

2.3

±

0.66

3.42

±

0.71

3.08

±

0.52

<0.001

Triglyceride (mg/dL)§

85.8

±

5.88

66.9

±

4.45

113.15

±

8.01

106.63

±

11.98

<0.001

HDL-C (mg/dL)

65.2

±

1.55

66.6

±

2.45

58.65

±

1.98

63.75

±

3.63

0.037

LDL-C (mg/dL)§

114.9

±

3.05

124.7

±

5.85

128.54

±

4.36

127.25

±

11.46

0.055

Total-C (mg/dL)§

188.2

±

3.08

196.3

±

6.21

199.04

±

4.89

201.75

±

11.13

0.235

hs-CRP (mg/dL)§

0.58

±

0.14

0.38

±

0.06

1.74

±

0.73

2.85

±

0.75

<0.001

Means±S.E., tested by one-way ANOVA, §tested after log-transformed; P<0.05 indicates significant differences among the values for the same variable. BP: blood pressure, HOMA-IR: Homeostatic model assessment for insulin resistance, HDL-C: high density lipoprotein cholesterol, LDL-C: low density lipoprotein cholesterol, Total-C: total cholesterol, hs-CRP: high sensitivity C-reactive protein.

2.5 It is unclear that the intra-assay and inter-assay variations were for adiponectin or BDNF assay? The authors should provide this information for both assays.

   Answer) In accordance with your comment, we provided the information for both assays.

Plasma adiponectin concentrations were measured using an enzyme immunoassay (R&D Systems, Minneapolis, USA, DRP300). The assays were performed in duplicate using a microplate absorbance reader (Bio-Rad Laboratories, Hercules, CA, USA) set to 450nm (intra-assay and inter-assay variations were less than 8% and 5%, respectively). Serum levels of mature-BDNF were measured using with an ELISA Kit (Aviscera bioscience, Santa Clara, CA, USA, SK00752-01). The assay was performed in duplicate using a microplate absorbance reader (Bio-Rad Laboratories, Hercules, CA, USA) set to 450nm (intra-assay and inter-assay variations were less than 10% and 6%, respectively).”

2.6 Ideally is to use primary adipocytes for the in vitro experiments, to align with the in vivo human study. Additionally, because both adiponectin and BDNF have been reported to induce brown adipocyte function, it is ideal to study brown adipocytes in addition to 3T3-L1 white adipocytes. Such consideration needs to be included into the manuscript.

Answer) Thank you very much for your comments for improving this manuscript. As you commented, it would be nice if the experiments had been performed on primary adipocytes. We used 3T3-L1 cells based on the previous reports that 3T3-L1 preadipocyte have been used for observing phenotype changes related to fat browning and adipogenesis [10.3389/fphys.2019.01380, 10.1038/s41598-018-20821-3, 10.1186/s12986-019-0361-8]. We additionally mentioned the necessity of using primary adipocytes in the in vitro experiments to more clearly support the study conclusion for the role of adiponectin and BDNF in brown adipocyte functions at the discussion part.

The study used insulin in the cell culture medium. Is this a standard procedure? Does insulin have crossover action as BDNF (a growth factor)?

Answer) We used insulin in the cell culture medium according to the standard protocol provided by ATCC. As mentioned in the manuscript, IBMX, dexamethasone, and insulin were added to the 3T3-L1 differentiation medium following the standard protocol. Insulin binds to the adipocyte membrane and activates enzymes related to promotion of differentiation [10.1038/cddis.2016.455, 10.1038/srep42104]. However, insulin doesn’t have crossover action with BDNF, because the period of each treatment was not overlapped as shown in method section and Fig. 1.

 2.7 The dosages of Acrp30 (20 ng/ml) and DHF (20 μM) need to be justified. BDNF serum concentration uses unit ng/mL. The authors should use one set of units in one manuscript.

Answer) As you advised, we unified the units of the treatment. We changed “20 μM of 7,8-DHF concentration” into “ 5 μg/mL of 7,8-DHF” in the manuscript

2.9 Does the Western blot antibody for BDNF label mature BDNF only?

 Answer) The authors are sorry again for making the reviewer confused. It was wrong , so we deleted it from the method section.

2.11 The statistical analysis method listed in this section (t-test) is not consistent with the method list in figure legend (one-way ANOVA). Need to be consistent.

   Answer) The authors are sorry for making the reviewer confused. We revised the explanation more clearly.

“Differences of parameters among the three groups were tested by one-way analysis of variance (unadjusted) (P0), and general linear model by Bonfferoni correction with adjustment for age, smoking, drinking, menopausal status and total calorie intake (P1). Differences of parameters between the two groups were tested by the Student t-test method.”

  1. Discussion:

BDNF levels were not different among three cohorts of women. Are BDNF expression (gene and protein) and circulating levels changed by body fat in men and women and male and female animal models? (ref: DOI: 10.1016/j.physbeh.2014.03.028)

Answer) As you commented, previous studies reported the relationship between BDNF and adiposity, and suggested circulating BDNF as a monitoring marker for assessing the step of visceral adiposity. For example, Plasma BDNF levels were correlated positively with BMI and fat mass in women, but which was not observed in men [38]. Serum BDNF level was increased and associated with obesity in women with newly diagnosed type 2 diabetes mellitus [10.1016/j.metabol.2006.02.012]. In addition, the animal experiments also demonstrated that reduced expression of BDNF during high fat diet increased a drive to eat, thereby contributing to diet induced obesity in male rat, but it was not observed in female rat [10.1016/j.physbeh.2014.03.028]. The experiment also shows that the expression of BDNF mRNA in the ventromedial nucleus of the hypothalamus was stable in female rat even when energy homeostatsis was disturbed. It may suggest sex-distinct regulation of central BDNF expression by diet and energy status [10.1016/j.physbeh.2014.03.028]. However, in our study women, mature-BDNF levels were not significantly different according to obesity status. We assumed that the discrepancy between our results and those from previous studies may be related to the degree of obesity and the baseline health status. Our study used the Asian criteria for obesity (BMI ³ 25 kg/m2) instead of the Western criteria (BMI ³ 30 kg/m2), and included relatively healthy subjects without diagnosed disease. Therefore, our study subjects were less obese and showed not severe metabolic abnormality than compared with obese people by the Western criteria. In the future, we included more obese people which can be applied by the Western criteria

Line 323: The author discussed adiponectin level affected by aging and presumably low estrogen hormone. Is BDNF regulated by aging and/or estrogen?

Answer) In accordance with your comments, we added explanation for the relationship between BDNF and aging/and or estrogen in the discussion part.

“Previous studies have reported that circulating BDNF levels were decreased significantly with age being increased [61-63]. However, a clinical study showed contradictory results according to age groups [64]. BDNF concentration were positively correlated with age in the middle aged (< 65 years), but negatively correlated in the elderly people (<=65 years). Circulating BDNF levels were also reported to be related to adiposity, and suggested as a monitoring marker for assessing the step of visceral adiposity [38,51]. For example, plasma BDNF levels were correlated positively with BMI and fat mass in women, but which was not observed in men [10.1371/journal.pone.0010099]. Serum BDNF level was increased and associated with obesity in women with newly diagnosed type 2 diabetes mellitus [10.1016/j.metabol.2006.02.012]. The animal experiments demonstrated that reduced expression of BDNF during high fat diet increased a drive to eat, thereby contributing to diet induced obesity in male rat, but it was not observed in female rat [10.1016/j.physbeh.2014.03.028]. It also shows that the expression of BDNF mRNA in the ventromedial nucleus of the hypothalamus was stable in female rat than in male rats even when energy homeostatsis was disturbed. It may suggest sex-distinct regulation of central BDNF expression by diet and energy status [10.1016/j.physbeh.2014.03.028].  

Line 346-347: “BDNF level is considerably linked to the alteration of metabolic risk factors, leading to metabolic disorder such as obesity, diabetes, CVD and CNS diseases”. BDNF has been suggested to suppress appetite and increase energy expenditure, thus reduce metabolic risks, which is opposite to this statement. How do the authors reconcile inconsistent literature? How does aging interact with BDNF’s effects in metabolic regulation? Again, the ages of woman subjects in the human study should be included in the manuscript.

Answer) Again, the authors are sorry for making the reviewer confused. We revised the sentences clearly and added more explanation for readers’ understanding. In accordance with your comment, we included the ages of study participants.

Round 2

Reviewer 2 Report

The authors responded to reviewers' comments and suggestions and revised accordingly. 

This manuscript is a resubmission of an earlier submission. The following is a list of the peer review reports and author responses from that submission.